# Utility of Protein Markers in COVID-19 Patients

**DOI:** 10.3390/ijms26020653

**Published:** 2025-01-14

**Authors:** López-Biedma Alicia, Onieva-García María Ángeles, Martín-García Desirée, Redondo Maximino, García-Aranda Marilina

**Affiliations:** 1Research and Innovation Unit, Hospital Costa del Sol, Autovía A-7 km 187, 29603 Marbella, Spain; alicia.lopez@ibima.eu (L.-B.A.); desiree.martin@ibima.eu (M.-G.D.); marilina.garcia@ibima.eu (G.-A.M.); 2Instituto de Investigación Biomédica de Málaga-Plataforma BIONAND (IBIMA-BIONAND), Severo Ochoa, 35, 29590 Malaga, Spain; 3Preventive Medicine and Public Health Unit, Hospital Universitario Reina Sofia, 14004 Cordoba, Spain; mariaa.onieva@gmail.com; 4Preventive Medicine and Public Health Research Group, Maimonides Biomedical Research Institute of Cordoba (IMIBIC), 14004 Cordoba, Spain; 5Department of Medical and Surgical Sciences, University of Cordoba, 14004 Cordoba, Spain; 6Red de Investigación en Servicios de Salud en Enfermedades Crónicas (REDISSEC) and Red de Investigación en Cronicidad, Atención Primaria y Promoción de la Salud (RICAPPS), Instituto de Investigación Biomédica de Málaga (IBIMA), 29590 Malaga, Spain; 7Surgical Specialties, Biochemistry and Immunology Department, Faculty of Medicine, University of Málaga, 29010 Malaga, Spain

**Keywords:** biomarkers, COVID-19, proteomic, treatment, risk

## Abstract

COVID-19 has been a challenge at the healthcare level not only in the early stages of the pandemic, but also in the subsequent appearance of long-term COVID-19. Several investigations have attempted to identify proteomic biomarkers in an attempt to improve clinical care, guide treatment and predict possible patient outcomes. Proteins such as C-reactive protein (CRP) or interleukin 6 (IL-6) are clear markers of severe disease, but many others have been proposed that could help in risk stratification and in the prediction of specific complications. This review aims to bring together the most relevant studies in this regard, providing information to identify the most notable biomarkers in relation to COVID-19 found to date.

## 1. Introduction

In modern medicine, biomarkers have become fundamental tools for improving the diagnosis, prognosis, and treatment of various diseases. Their ability to accurately reflect biological processes, whether normal, pathological, or in response to therapeutic interventions, positions them as key elements in clinical practice. From the early identification of diseases to the personalization of treatments, biomarkers have transformed the predominant approach to healthcare, enabling a more precise and evidence-based decision-making process. Furthermore, their integration with advanced technologies, such as proteomics and genomics, has significantly expanded their potential, opening new opportunities for the development of innovative therapies and more effective prevention strategies.

In December 2019, an outbreak of viral pneumonia caused by a novel coronavirus, later named SARS-CoV-2 (Severe Acute Respiratory Syndrome Coronavirus 2), was identified in the city of Wuhan, China. This disease, now known as Coronavirus Disease 2019 (COVID-19), rapidly spread across the globe, triggering an unprecedented pandemic that overwhelmed healthcare systems and created a global public health crisis [1]. COVID-19 demonstrated a highly variable clinical spectrum, ranging from asymptomatic or mild cases with symptoms like fever, fatigue, and cough to severe respiratory failure with multi-organ involvement [2].

SARS-CoV-2 primarily spreads through airborne transmission, infecting epithelial cells via binding to angiotensin-converting enzyme-2 (ACE2) receptors. These receptors are widely distributed, not only in lung tissue but also in the vascular endothelium, intestines, brain, and organs like the liver and kidneys [3]. This broad distribution explains why COVID-19 is not just a respiratory illness but a multisystemic disease capable of causing severe complications. Among its most concerning effects is the overactivation of the immune response, known as the “cytokine storm,” characterized by excessive cytokine production and an uncontrolled inflammatory response that worsens the disease and increases mortality risk [2].

On a molecular level, SARS-CoV-2, an enveloped, single-stranded ribonucleic acid (ssRNA) virus, employs specific proteins to facilitate its entry into host cells and replication. The spike (S) protein, for instance, plays a critical role by interacting with the ACE2 receptor, initiating the infection process. This cascade disrupts the immune system and other molecular pathways, leading to the dysregulation of the renin–angiotensin–aldosterone system (RAAS), epithelial and endothelial damage, and fibrosis in severe cases. Such complexity increases susceptibility to severe multisystemic conditions, including renal, hematologic, cardiovascular, gastrointestinal, hepatic, or neurological damage, often with a poor prognosis [4,5].

Rapid identification and containment of SARS-CoV-2 has been one of the greatest challenges since the onset of the pandemic. Due to its high transmissibility and the emergence of variants with mutations in the S protein, early detection became a critical priority. Reverse transcription polymerase chain reaction (RT-PCR) tests have been the gold standard for COVID-19 diagnosis due to their accuracy and reliability [6,7,8]. However, RT-PCR has limitations, such as the need for specialized personnel, expensive equipment, and susceptibility to false-negative or false-positive results under certain conditions. Additionally, it has limited utility in detecting infections at very early or late stages [9]. These challenges led to the rapid development of alternative diagnostic tools, including antigen-based tests and isothermal amplification techniques, which formed the basis for rapid tests and point-of-care devices aimed at controlling virus transmission [10].

As the pandemic evolved, long-term COVID-19, also known as post-COVID-19 syndrome or post-acute sequelae of SARS-CoV-2 infection (PASC), emerged as a significant concern. This condition affects individuals of all ages, regardless of the severity of their initial symptoms [11,12]. Long-term COVID-19 is characterized by persistent or new symptoms lasting more than two months, without an identifiable cause, and appearing at least three months after the initial infection [12]. It is estimated that 10% to 20% of patients with SARS-CoV-2 develop long-term COVID-19, with symptoms ranging from fatigue, brain fog, and shortness of breath to over 200 possible manifestations [11,13].

Since 2020, numerous studies have documented the long-term complications of COVID-19 in patients recovering from the acute phase. Between 10% and 30% of these patients have shown conditions involving multiple systems, including pulmonary, cardiovascular, renal, immunological, metabolic, and neurological impairments [14,15]. Even children with mild or asymptomatic cases have experienced persistent symptoms such as fever and fatigue [16]. With no specific cure available, treatment for long-term COVID-19 relies on a multidisciplinary approach focused on symptom management.

Despite the World Health Organization declaring the end of the global health emergency in May 2023, the global health burden continues to grow due to periodic resurgences in infections and the effects of long-term COVID-19. This highlights the urgent need to identify biomarkers that aid in early diagnosis, improve risk stratification, and optimize management of affected individuals. Biomarkers are especially critical for detecting early-stage infections and identifying asymptomatic carriers, who play a significant role in virus transmission, as well as for understanding and addressing long-term complications.

Large-scale “omics” approaches, including genomics, transcriptomics, and proteomics, have opened new avenues in biomarker discovery [17,18,19]. Among these, proteomics offers unique insights by characterizing the molecular alterations induced by COVID-19, providing crucial information about the interactions between SARS-CoV-2 and its host. Proteomic analyses have revealed dynamic changes in protein expression during infection and have identified differences between mild, severe, and critical cases. These findings have contributed to the identification of biomarkers that indicate disease severity, prognosis and potential therapeutic targets [20,21,22,23].

In this article, we aim to explore the existing literature on the use of proteomic biomarkers to assess risk and guide treatment in COVID-19. By summarizing key findings, we hope to highlight the potential of proteomics in advancing the understanding and management of this complex disease.

## 2. Objectives and Methods

For this review, we conducted a comprehensive search using the NCBI, PubMed, Web of Science, and Scopus databases, as well as the truthful information pages of the U.S. Centers for Disease Control and Prevention (CDC) and World Health Organization (WHO). To ensure a comprehensive review of the literature, we employed for the search the terms “proteomics”, “proteomic biomarkers”, and “LC-MS/MS”, along with the acronyms “SARS-CoV-2/COVID-19” and “long-term COVID-19/post-COVID-19”. We focused exclusively on peer-reviewed research articles published in English in international journals up to 31 October 2024. In addition, relevant studies were included from the reference list of eligible articles. We thoroughly reviewed the full texts of the selected articles, removing duplicate or redundant information. Our primary objective was to focus on studies that identify key proteomic biomarkers for disease prognosis and treatment.

## 3. Importance of Biomarkers in COVID-19

A biomarker is a measurable indicator that reflects a specific biological state, playing a crucial role in assessing the risk of developing a disease, as well as its presence or progression in the body [24]. Historically, biomarkers were primarily based on physiological metrics or physical traits. However, with the advent of molecular analysis, especially in diseases such as COVID-19, the application of biomarkers has undergone a revolution. Today, biomarkers are essential tools for understanding individual susceptibility, predicting disease progression, and evaluating responses to specific treatments.

Biomarkers can be categorized by their applications, although a single marker can serve multiple purposes when supported by specific evidence [19,24]. Diagnostic biomarkers help identify or confirm diseases and classify subtypes, as seen in cancer, where molecular features are now preferred over traditional organ-based criteria. Monitoring biomarkers assesses the status of a disease or the impact of medical and environmental agents. If a marker changes following exposure, it is classified as pharmacodynamic. Predictive biomarkers anticipate treatment responses, while prognostic biomarkers estimate the risk of clinical events, disease progression, or recurrence. These are distinct from susceptibility biomarkers, which identify the likelihood of developing a disease.

In the context of COVID-19, biomarkers have played a fundamental role in early detection, diagnosis of disease severity, treatment monitoring, and prognosis prediction [25]. Additionally, they have facilitated the classification of patients based on their molecular profiles, enabling the identification of those at higher risk of severe complications. Global research efforts have sought reliable biomarkers to optimize patient stratification and clinical management. Some biomarkers not only indicate the presence of infection but also guide pharmacological decisions and serve as potential therapeutic targets in severe cases [26,27].

Understanding how biomarkers evolve throughout the course of COVID-19 is essential for personalized and effective clinical management. These advancements emphasize the importance of biomarkers, not only in current treatments but also in the development of future therapies and medical strategies.

The choice of biological samples for proteomic analysis is crucial in addressing clinical and research challenges. While invasive samples provide detailed information on systemic and local responses to SARS-CoV-2, non-invasive samples offer ease of collection and a lower risk of exposure. Both types of samples are complementary and vital for advancing our understanding of COVID-19 pathophysiology, as well as developing new strategies for diagnosis, prognosis, and treatment (Table 1). Selecting the appropriate biological sample is essential, as it impacts the detection of specific biomarkers related to disease progression, inflammation, and systemic complications.

Blood is the most common source of proteomic biomarkers in patients with COVID-19. It enables the identification of proteins related to inflammatory responses, such as cytokines, Interleukin-6 (IL-6), Tumor necrosis factor α (TNF-α), and acute phase proteins, which correlate with the cytokine storm observed in severe cases. Markers of endothelial damage have also been detected, such as elevated levels of selectin E and von Willebrand factor (vWF), which are associated with vascular dysfunction and thrombosis in patients with severe COVID-19, and coagulation markers (D-dimer and fibrinogen) related to the risk of thromboembolic events and disseminated intravascular coagulopathy [28,29,30].

Cerebrospinal fluid, although less common, has been used in cases of COVID-19 with neurological complications to identify biomarkers related to encephalitis or central nervous system damage. In some cases, the presence of SARS-CoV-2 RNA has been detected, indicating invasion of the central nervous system [31]. Cytokines such as IL-6 and chemokines such as C-X-C motif chemokine 10 (CXCL10 or IP10) have been detected, reflecting inflammation in the central nervous system (CNS), as well as elevated levels of neurofilament light protein (NFL) and S100B protein, indicative of axonal damage and blood–brain barrier dysfunction. Autoantibodies against CNS structures have also been identified, suggesting an autoimmune component in the severe neurological manifestations of COVID-19 [32,33].

Lung tissues have provided key information about COVID-19 lung pathology, including proteomic biomarkers reflecting severe alveolar damage and fibrotic responses. The presence of reduced surfactant and elevated levels of proteins such as surfactant protein A (SP-A) and surfactant protein D (SP-D) indicate alveolar dysfunction, while the presence of collagen types I and III and elevated levels of Transforming Growth Factor β (TGF-β) have been associated with pulmonary fibrosis in severe cases [34,35]. Other local inflammation proteins detected, such as Interleukin 8 (IL-8), would be involved in attracting neutrophils and macrophages to lung tissue. Finally, fibrin deposits and elevated levels of thrombomodulin reflect thrombotic microangiopathy in pulmonary capillaries [30,36].

Bronchoalveolar lavage (BAL) provides a direct window into the immunological and molecular environment of the lungs, allowing for the identification of key biomarkers in patients with COVID-19, especially in severe cases. BAL has been one of the most reliable samples for the detection of viral RNA in the lower respiratory tract, being crucial to confirming infections in patients with negative tests in nasopharyngeal swabs but with high clinical suspicion [37]. BAL analysis has revealed elevated levels of IL-6, Interleukin 1β (IL-1β), Tumor Necrosis Factor (TNF-α), and chemokines such as CXCL10 and Chemokine (C-C motif) ligand 2 (CCL2), reflecting the activation of the immune response in the lung and its relationship with the cytokine storm. Proteins such as C-reactive protein (CRP) and pulmonary surfactants have been found, indicating lung damage and disease progression. On the other hand, flow cytometry and transcriptomic studies in BAL have identified an increase in activated inflammatory monocytes and macrophages, accompanied by exhausted T cells, suggesting local immune dysfunction in the lungs. BAL has also revealed biomarkers related to microthrombosis, such as fibrin fragments, which may be indicative of localized intravascular coagulopathy in lung tissue [37,38].

Aqueous humor, although not commonly used, has demonstrated the ability to infect intraocular tissues, with viral RNA detected in this sample in selected cases [39]. Elevated levels of proinflammatory cytokines, such as IL-6, IL-8, and TNF-α, have been identified, reflecting local inflammation and its possible connection to the systemic inflammatory state of COVID-19 patients. Likewise, the presence of proteins related to intraocular endothelial damage has been identified, which may be linked to SARS-CoV-2-induced microangiopathy [39].

Nasopharyngeal and oropharyngeal swabs are the most used samples for initial diagnosis by RT-PCR, but they have also allowed for the identification of viral and host proteins associated with the progression of the infection [40]. Structural components such as the N protein (nucleocapsid) and the S protein (spike) have been detected in these samples, which correlate with viral load, the presence of IL-6 and TNF-α associated with local inflammation in the upper respiratory tract, or host proteomic markers, such as peroxiredoxin 1 (PRDX1) and annexin A1 (ANXA1), linked to the inflammatory response and disease severity [40].

Saliva has emerged as a promising alternative for the diagnosis and monitoring of COVID-19 [41]. It contains proteomic biomarkers related to local and systemic immune responses, such as detectable levels of SARS-CoV-2-specific Immunoglobulin A (IgA) or elevated levels of IL-1β and IL-8, indicative of inflammation in the oral mucosa [42,43].

Regarding urine, although not widely used, it may reflect systemic changes induced by COVID-19, including biomarkers of inflammation and kidney damage, especially in cases with renal complications. Increased levels of N-acetyl-beta-D-glucosaminidase (NAG) and neutrophil gelatinase-associated lipocalin (NGAL), indicators of renal tubular injury, have been detected, as well as the presence of cytokines such as IL-6 in urine, which correlates with systemic inflammation [44,45].

Finally, stool samples have been key to studying viral persistence in the gastrointestinal tract and the effects of COVID-19 on the intestinal microbiota. They also contain proteomic biomarkers associated with intestinal inflammation [46]. The presence of viral RNA has been detected in patients with negative nasopharyngeal tests, suggesting an intestinal reservoir of the virus. An increase in IL-1β and TNF-α has also been detected in fecal content, indicative of intestinal inflammation, detection of fragments of the N protein in the stool of patients with gastrointestinal symptoms, and alterations in the abundance of certain bacterial proteins associated with an imbalance in the intestinal microbiota [46,47,48].

### 3.1. Towards Multimodal Biomarkers in the Diagnosis and Clinical Management of COVID-19

The search for biomarkers in the context of COVID-19 has been critical in understanding the disease’s pathophysiology, stratifying patient severity, and optimizing therapeutic strategies. Over time, various types of molecules have been identified as potential biomarkers, including nucleic acids, metabolites, lipids, and proteins. In recent years, integrating these biomarkers into multimodal platforms has shown great promise in offering a more comprehensive understanding of the disease, thereby enhancing diagnostic and prognostic accuracy.

Nucleic acids have played a pivotal role in the diagnosis of COVID-19, primarily through the direct detection of SARS-CoV-2 viral RNA. The RT-PCR technique, widely regarded as the gold standard, enables the precise detection of viral genome fragments in respiratory samples, such as nasopharyngeal swabs and saliva. Additionally, plasma viral RNA levels have been proposed as biomarkers for monitoring viral load and assessing disease progression [49]. Moreover, microRNAs (miRNAs), small molecules that regulate gene expression, have emerged as promising biomarkers in COVID-19. Alterations in miRNAs, particularly those involved in inflammation and immune response, like miR-146a, have been linked to disease severity, further enhancing our understanding of the virus’s impact [50].

The study of metabolites and lipids has also revealed significant alterations in patients with COVID-19. Metabolites such as lactate and pyruvate, which are involved in energy metabolism and hypoxia, have shown imbalances, particularly in critically ill patients [51]. Similarly, lipid profiling has demonstrated significant changes, such as a reduction in phospholipids and polyunsaturated fatty acids in plasma. These alterations may indicate heightened systemic inflammation and oxidative stress, both of which are central to the disease’s pathophysiology [52].

Protein biomarkers have been extensively studied in the context of COVID-19, especially in serum and other biological fluids. Molecules like cytokines (e.g., IL-6, TNF-α) have been associated with the cytokine storm observed in severe cases. Additionally, proteins such as C-reactive protein (CRP) and D-dimer have proven useful as predictors of systemic inflammation and the risk of thrombotic events [1]. These biomarkers are instrumental in assessing the severity of the disease and guiding clinical decisions.

The integration of these diverse biomarkers into multimodal platforms has the potential to revolutionize COVID-19 diagnosis and management. These platforms combine data from nucleic acids, metabolites, proteins, and lipids, using artificial intelligence and machine learning algorithms to create predictive models. These models not only improve diagnostic accuracy but also offer enhanced prognostic capabilities, which is crucial in managing complex diseases like COVID-19, where multiple pathophysiological pathways are involved.

One of the key advantages of multimodal biomarkers is their ability to overcome the limitations of single-biomarker approaches. While individual biomarkers, such as a specific protein or metabolite, can provide valuable insights, they often fail to capture the complexity of multifactorial diseases like COVID-19. Multimodal approaches, on the other hand, incorporate multiple layers of molecular data, along with demographic and clinical information, offering a more comprehensive understanding of the disease. This holistic view enables the development of more robust predictive models, which are essential for stratifying patients based on their risk, personalizing treatments according to individual molecular signatures, and predicting therapeutic responses with greater precision.

In the context of COVID-19, multimodal biomarker approaches could significantly improve clinical management by identifying patient subgroups that may benefit from specific therapies, minimizing the risk of severe complications and optimizing the use of medical resources in high-demand scenarios. By offering a more complete picture of the disease, these innovative approaches hold the potential to transform the way COVID-19 is diagnosed, monitored, and treated, ultimately leading to more effective and personalized patient care.

### 3.2. Technological Advancements in Proteomics

The growing demand for COVID-19-specific proteomic biomarkers has spurred significant advancements in proteomics technologies. Cutting-edge techniques, such as mass spectrometry (MS), reverse-phase protein arrays, antibody/antigen arrays, proximity extension assays, and aptamer assays, are now being employed to analyze protein changes related to viral infections (Figure 1). These technologies enable a detailed and accelerated analysis of infection-associated proteins, crucial for responding to the pandemic and improving future intervention and prevention strategies.

Mass spectrometry (MS) is the most widely used proteomics technology, and there has been rapid progress in MS-based techniques, such as matrix-assisted laser desorption/ionization (MALDI-TOF MS), electrospray ionization (ESI-MS), liquid chromatography coupled with mass spectrometry (LC-MS/MS), Orbitrap, and Fourier transform MS. MALDI-MS, known for its high throughput, is ideal for analyzing relatively simple peptides, while LC-MS/MS is used to study more complex peptide mixtures [53]. On the other hand, ESI-MS, which allows for the identification of proteins by digesting samples and ionizing peptides, is particularly suited for exploratory research, although it faces challenges in high-throughput applications and analyzing post-translational modifications (PTMs) [54].

## 4. Biomarkers in COVID-19

### 4.1. Classic Biomarkers in COVID-19

The most widely used and validated protein biomarkers for reliable stratification of patients according to expected outcome include CRP and IL-6, both of which are correlated with the cytokine storm induced by SARS-CoV-2 [1,27]. Other hematological biomarkers, such as D-dimer and ferritin, are markers of inflammation and show high sensitivity but low specificity, since their levels are elevated in many different pathological conditions. Their relationship with the severity and mortality of COVID-19 is detailed in depth in the review of Rizzi et al. [27].

CRP is a cytokine-mediated acute phase reactant and determination of plasma CRP levels and serves as a useful tool to identify those patients who require urgent care and closer clinical follow-up. Its levels serve both as an early indicator to classify the severity and to predict the evolution of the disease [55,56,57]. In the progression phase, patients with COVID-19 present higher levels of CRP than those in a stable state, and its measurement at 7 days of hospitalization has been found to be a reliable marker to evaluate the response to treatment in patients with moderate or severe COVID-19 [58]. After a systematic review, Nazemi et al. [59] even propose a prognostic threshold of 75 mg/dl for serum CRP levels, above which COVID-19 patients would present severe disease.

Like CRP, IL-6 is found in very low circulating levels under normal conditions, but experiences a sustained increase in acute situations and is valuable not only upon admission of the patient as a predictor of negative outcomes, but also throughout the hospitalization to guide therapeutic decisions [60,61]. As a modulator of the immune system, IL-6 influences both the innate and adaptive immune responses. On the one hand, it can alter the activity of CD8+ T cells and natural killer (NK) cells, which reduces the ability to defend against the virus. On the other hand, it can interfere with the adaptive immune response by promoting the differentiation of B cells into antibody-producing plasma cells and by regulating the differentiation of CD4 T cells into Th2 and Th17 lymphocytes. In relation to COVID-19, its serum levels increase significantly in non-survivors compared to survivors and in comparison with severe versus non-severe disease, being associated with vasculitic processes underlying organ damage in severely ill patients, such as edema, acute respiratory distress syndrome, cardiovascular damage, and brain lesions [56].

It seems sensible, therefore, that treatment strategies should be directed at reducing IL-6 production, and recent studies point to certain drugs such as auranofin, which showed protective action in vivo through the reduction in viral replication, IL-6 production and inflammation in the lungs [62], and siltuximab or tocilizumab, both of which are monoclonal antibodies specifically directed against IL-6 [63,64].

As previously reported by Malik et al. [26], several studies have shown that severe or fatal cases of COVID-19 are associated, in addition to with CRP and IL-6, with elevated white blood cell counts, blood urea nitrogen, creatinine, liver and kidney function markers, and lower lymphocyte and platelet counts, as well as albumin levels compared to milder cases whose outcome is survival. In their systematic review, they also found that lymphopenia, thrombocytopenia, and elevated levels of D-dimer, procalcitonin (PCT), lactate dehydrogenase (LDH), alanine aminotransferase (ALT), aspartate aminotransferase (AST), and creatinine kinase (CK) were also associated with poor outcomes in hospitalized patients. Ponti et al. [56] agree on several of these parameters, distinguishing hematological (lymphocyte count, neutrophil count, neutrophil-lymphocyte ratio (NLR)), inflammatory (CRP, erythrocyte sedimentation rate (ESR), PCT), immunological (IL-6), and biochemical biomarkers (D-dimer, troponin, CK, AST), especially those related to coagulation cascades in disseminated intravascular coagulation (DIC) and acute respiratory distress syndrome (ARDS).

Patel et al. [65] found proteomic differences between COVID-19-positive patients (mild, severe, and critical) and controls, performing a blood protein profile with inflammation, autoimmune, cardiovascular, and neurological panels. Of the 368 proteins measured per individual, they observed that more than 75% were significantly altered in COVID-19 cases and identified six proteins (IL6, CKAP4, Gal-9, IL-1ra, LILRB4, and programmed cell death protein ligand-1 (PD-L1) associated with disease severity.

Onoja et al. [66] evaluated a collaborative panel comprising the most commonly mentioned COVID-19 metabolomic biomarkers in the literature between 2020 and 2023 and found that the best performing panel in the independent dataset included nine biomarkers: lactic acid, glutamate, aspartate, phenylalanine, β-alanine, ornithine, arachidonic acid, choline, and hypoxanthine. In the case of alanine, in the most recent literature, we found that levels were significantly different between COVID-19 patients with infections of different severities, considering its reduction as a negative prognostic factor in such patients [67].

To differentiate between patients who required invasive mechanical ventilation (IMV) versus those who, even with acute respiratory failure, did not require it (NIV), a study evaluated serum levels of matrix metalloproteinase 7 (MMP-7) and T-cell-related molecules as potential biomarkers. The first group showed higher levels of sMMP-7, sPD-L1, and mucin domain-containing T-cell immunoglobulin-3 (sTIM-3). sMMP-7 had the highest sensitivity (78%) and specificity (76%) and remained elevated eight months after acute infection, suggesting it as a marker of persistent post-COVID-19 lung injury [68].

We have already mentioned that COVID-19 disease is not limited to a respiratory pathology but is multisystemic. Therefore, beyond the general approach to distinguish between patients with or without risk of severe illness from COVID-19, some studies have focused on certain complications derived from the disease, such as the appearance of atrioventricular block (AVB), in which several differentially expressed proteins in plasma were identified (serum amyloid A protein, tetranectin, and neutrophil defensin 3) [69] or cases of secondary hemorrhagic lymphohistiocytosis (sHLH), a hyperinflammatory disorder related to the aforementioned cytokine storm which can complicate infectious processes. In the latter case, Canny et al. [70] identified three biomarkers associated with sHLH (soluble PD-L1, TNF-R1 and IL-18BP), in addition to the syntaxin pathway, involved in promoting viral infection.

In a large-scale study of 3281 hospitalized COVID-19 patients, it was found that 82% of these patients had elevated levels of one of the three biomarkers that reflect the pathobiological axes of myocardial injury, coagulation, and inflammation (cardiac troponin, D-dimer and CRP, respectively). These levels were directly related to mortality and stepwise increases in the risk of adverse events. In contrast, patients without these elevated biomarkers had a lower risk of critical illness and in-hospital mortality [71].

Keeping in mind the aforementioned organic morbidity, Bandyopadhyay’s group performed an analysis of three proteomic studies to identify potential protein signatures for organic dysfunctions as a complication of COVID-19 [72]. They found six proteins as potential biomarkers that demonstrated a consistent association with organic disorders: Vitamin K-dependent protein S and Antithrombin-III for signaling thrombotic disorder, voltage-dependent anion-selective channel 1 for neurological disorder, Filamin-A for cardiovascular system-related diseases, and Peptidyl-prolyl cis-trans isomerase A and Peptidyl-prolyl cis-trans isomerase FKBP1A for gastrointestinal disorders.

In another study, the immune response profile was evaluated using a total of 71 biomarkers in serum from control groups and asymptomatic, non-hospitalized and hospitalized SARS-CoV-2 infected patients (severe cases). Among the angiogenesis markers, epidermal growth factor (EGF), IL-8, hepatocyte growth factor (HGF), Heparin-binding EGF-like growth factor (HB-EGF), Vascular endothelial growth factor C (VEGF-C), and Vascular endothelial growth factor A (VEGF-A) were elevated more frequently in severe cases compared to the other groups. The same occurred for cardiovascular disease biomarkers such as D-dimer, Growth Differentiation Factor 15 (GDF-15), and Soluble Intercellular Adhesion Molecule 1 (sICAM-1), while there were no differences in these biomarkers between the control and asymptomatic groups. They also identified important differences in cytokines and chemokines according to the clinical course. Severe cases had elevated levels of IL-6, IP-10, Macrophage colony stimulating factor (M-CSF), Macrophage derived chemokine (MDC), and Macrophage Inflammatory protein 1 (MIP-1) compared to the control group [73]. Other studies support the aforementioned role of many of these markers, such as for GDF-15 [74,75,76], ICAM-1 [77,78,79], M-CSF [80], or MIP-1 [79,81].

### 4.2. Emerging Biomarkers in COVID-19

As we have already mentioned, biomarkers such as CRP, D-dimer, and IL-6 are clear predictive markers and have been used in clinical care for some time, since they are significantly expressed in patients hospitalized with COVID-19. However, in the last year, the list of investigated biological signatures has expanded (Table 2), which may help to further refine patient management, prognosis, and resource allocation.

Many of the recent markers found coincide with the aforementioned classic markers, which have the problem of not being specific for COVID-19 disease. However, we can find some more specific ones. Such is the case of the study of Harriott and Ryan [89], who observed that proteins such as TNFβ, IL-6, IL-8, IL-12, CXCL10, syndecan-1 (SYND1), and EN-RAGE were positively regulated in cases of severe COVID-19, after comparing the expression of proteins in the blood plasma of patients with different severities (mild, moderate, or severe). Nevertheless, they identified new proteins not previously associated and expressed at significantly higher levels in subjects with severe COVID-19, among which were a group of growth factors and associated proteins that included growth factor 5 (FGF-5), colony stimulating factor (CSF), ephrin type-A (EPHA2), transforming growth factor alpha (TGF-α), and beta-nerve growth factor (β-NGF). Of these growth markers, mesothelin (MSLN) was one of the most significantly elevated proteins. They also identified a subset of proteins significantly elevated in milder vs. severe COVID-19 disease (receptor activator for nuclear factor κβ (TRANCE/RANKL), Fas ligand (FASLG), X-prolyl aminopeptidase 2 (XPNPEP2), and langerin (CD207)), representing “efficient” disease response and better prognoses.

Meanwhile, another study compared proteomic responses in the plasma of patients with COVID-19 and sepsis, identifying 42 common proteins associated with infection, although with higher levels in sepsis. However, they found three markers, E3 ubiquitin-protein ligase (TRIM21), Pleiotrophin (PTN), and Caspase-8 (CASP8), that differentiated COVID-19 from community-acquired pneumonia sepsis more accurately than standard clinical markers [91]. Similarly, Patel et al. [90] found that plasma proteome of critically ill COVID-19 patients was distinguishable from that of non-COVID-19 sepsis controls and healthy control subjects. In this case, they found an optimal nine-protein model of specific proteins with multi-system expression and correlated with hemoglobin, coagulation factors, hypertension, and high-flow nasal cannula intervention, i.e., (Platelet factor 4 variant (PF4V1), Nucleobindin 1 (NUCB1), CRK Like Proto-Oncogene, Adaptor Protein (CrkL), SerpinD1, Flap Structure-Specific Endonuclease 1 (Fen1), GATA-4, Proprotein convertase subtilisin/kexin type 1 inhibitor (ProSAAS), Parkinsonism associated deglycase (PARK7), and neuroepithelial cell transforming 1 (NET1)).

Finally, it is necessary to mention certain recent approaches that deserve to be considered. The study of Siwy et al. [92] proposed for the first time that SARS-CoV-2 infection very significantly increases the risk of mortality not only during the acute phase of the disease, but for a period of approximately one year, identifying 201 urinary peptides linked to such mortality. These peptides are albumin fragments, alpha-2-HS-glycoprotein, apolipoprotein AI, beta-2-microglobulin, CD99 antigen, various collagens, fibrinogen alpha, polymeric immunoglobulin receptor, sodium/potassium-transporting ATPase, and uromodulin that they integrated into a predictive classifier (DP201). Higher DP201 scores, together with age and BMI, significantly predicted death.

For their part, Viode et al. [84], through a comprehensive multicenter study, highlighted the importance of longitudinal sampling, as they identified changes in the plasma proteome during 28 days of hospitalization, which could provide more information than baseline admission values on the outcome of the disease. They also found biomarkers to distinguish between survival and fatal outcome (IL1RL1 and elastin, ELN) and an association of neutrophil extracellular traps (NET) and markers of cardiac damage in the most severe cases of COVID-19 with a fatal outcome.

Furthermore, a recent study has shown that mild and severe SARS-CoV-2 infections remain in the nasopharyngeal mucosa two years after infection with biomarkers that define each group, with ubiquitin-protein ligase (XIAP) and E3 ubiquitin-protein ligase (UBR4) being the best correlated with the severity of symptoms [93].

### 4.3. Biomarkers and Long-COVID-19

In relation to persistent or prolonged COVID-19, biomarkers such as taurine have recently been evaluated, whose increase in plasma during the transition to convalescence was associated with a reduction in the risk of developing said post-COVID-19 syndrome, opening the door to a possible supplementation with this amino acid after the acute phase of the disease [83]. On the other hand, a systematic review found that 113 biomarkers are significantly associated with long COVID-19, of which 38 (33.6%) were cytokines/chemokines, 24 (21.2%) biochemical markers, 20 (17.7%) vascular markers, 6 (5.3%) neurological markers, and 5 (4.4%) acute phase protein. Compared with healthy control or recovered patients without long-COVID-19 symptoms, 79 biomarkers were increased and 29 were decreased. Among these, elevated levels of IL-6, CRP, and tumor necrosis factor alpha (TNFα) for one or more months after SARS-CoV-2 infection were indicators that patients may experience long-COVID-19 symptoms [94].

These results are in line with another review in which the most frequently reported biomarkers were IL-6 and CRP, in addition to IL-10, interferon (IFN)-γ, α2 antiplasmin (α2AP), vWF, and regulatory T cells [95]. Therefore, the correlation between high levels of IL-6 after acute SARS-CoV-2 infection and the evolution to long-COVID-19 seems to be clear [96]. As already mentioned, MMP-7 also remained elevated eight months after acute infection, suggesting that it is a marker of persistent lung injury post-COVID-19 [68].

More recently, through a comprehensive proteomic analysis using plasma and pellet fractions, distinct proteomic profiles were identified between symptomatic and asymptomatic post-COVID-19 patients. In plasma, proteins involved in coagulation, immune response, oxidative stress, and metabolic regulation showed significant alterations in the symptomatic group, indicating an increased inflammatory state and prolonged immune activation. Proteomic analysis of the pellet fraction, which reflects cellular components such as red and white blood cells, revealed significant changes in proteins related to immune response, cellular stress, and iron metabolism [97].

Patel et al. [98] found significant differences even between patients with long-COVID-19 compared to a combined cohort of severely ill hospitalized COVID-19 patients and healthy controls, indicating an unique protein profile. In this study, 119 plasma biomarkers were identified to classify patients with PASC, among which an optimal set of nine proteins was determined, one of them significantly decreased (Frizzled related protein, FRZB) and eight of them significantly elevated (CXCL5, Adaptor-related protein complex 3 subunit sigma 2 (AP3S2), MYC-associated factor X (MAX), PDZ and LIM Domain 7 (PDLIM7), Ectodysplasin A Receptor (EDAR), Leukotriene A4 hydrolase (LTA4H), Calcium release activated channel regulator 2A (CRACR2A), and CXCL3). In addition, the 119 proteins were analyzed to identify expression patterns of organs and cell types. The digestive system had the highest number of significant proteins with altered expression, which helps explain some previously reported post-COVID-19 symptoms, such as changes in the gut microbiota or gastrointestinal and digestive problems [15,99]. However, once again, among the proteins studied, a change in expression was associated with different organs and cells involved, including leukocytes and platelets, which shows the multisystemic nature of long-term COVID-19.

## 5. Limitations in This Field and Future Lines of Work

There is no doubt about the importance of the study of proteomic biomarkers in the management of COVID-19 and the information it can give us in the prediction, diagnosis and prognosis of the disease and its subsequent consequences, such as the appearance of prolonged COVID-19. However, there are important limitations that should be taken into account to improve the information on the basis of which many decisions can be made in the clinical setting. One of them is the retrospective nature of most of the studies, given the eventuality of the pandemic and the emergency it represented. In fact, many of the results are postmortem after severe illness and without stratifying previous medical conditions that may confound biomarker analysis. The passage of time gives us the opportunity to develop long-term prospective studies, which would complement the information and take new variables into account.

A recent review that analyzes a large number of studies that describe possible biomarkers for long-COVID-19 shows that none of these biomarkers and no laboratory test or panel of biomarkers can differentiate long-COVID-19 from other pathological entities with adequate certainty and that there are many methodological limitations in the studies that evaluate them, such as selection biases, cohorts not being representative of the general population, low statistical power due to the small sample size, etc. [100]. In this regard, some markers have emerged that shed light on specificity for COVID-19 [89] and long-term COVID-19 [98], and efforts in new research should be directed in this direction.

Therefore, it should not be ignored that a biomarker cannot be translated into clinical practice for treatment guidance until it is shown to have a significant impact, since the level of almost all analytes can change in the face of a severe infection. Among the candidates, only a few biomarkers reliably predict a worse prognosis in patients with COVID-19, and even fewer molecules show the ability to predict responses to treatment. There are also significant variations in methods, cut-off points and results between studies, making definitive conclusions difficult. Selection bias and heterogeneity limit the robustness of meta-analyses, and sensitivity analyzes are necessary to avoid bias. Although many studies adjust their analyses, unmeasured confounders cannot be excluded [101]. Some studies have focused on this important aspect and have shown how intra-individual variability, methodological variability, or statistical problems can lead to the false discovery of biomarkers or the erroneous identification of affected pathways [66]. Thus, an important aspect to consider is the difference between studies when stratifying patients. Often, such stratification is oversimplified, without taking into account the different profiles within the same group, which can lead to divergent clinical results and treatment responses. Some trials divide patients as mild, severe, or critical disease severity, others as hospitalized/non-hospitalized, others as hospitalized in intensive care/wards, etc., so there is a lack of consensus on this aspect that complicates the extrapolation of results. In fact, the study by Verhoef et al. [102] revealed specific characteristics of severity and two distinct profiles with differential responses to treatment after analysis of protein biomarkers of hospitalized patients with COVID-19, i.e., a profile with higher levels of albumin and bicarbonate, and another profile with higher inflammatory markers. The latter had a longer mean length of hospital stay and mortality compared to those in the first profile; however, higher mortality was observed for patients in the first profile after treatment with glucocorticoids.

That said, the detection of the biomarkers must be validated. Biomarker validation is a process used to determine whether the performance of a biomarker is acceptable for its intended purpose. This process encompasses internal validation (using the data from which a biomarker was developed); external validation (using a completely independent dataset); analytical validation (determining sensitivity, specificity, accuracy, and precision); and clinical validation (confirming an association between the biomarker and the endpoint of interest and revealing the clinical utility of the biomarker) [95].

Surprisingly, despite the fact that the entire world has focused on studying COVID-19, the lack of evidence and reproducibility of the studies means that in many cases, biomarkers, although useful, are not sufficient to address all the complexities of the disease. The future study of biomarkers should be conducted taking into account all of the above-mentioned premises, since the better profiled all this knowledge is, the better the approach will be in clinical practice and its impact on patients with COVID-19 and long-COVID-19, the latter a true challenge for health systems once the pandemic seems to have stabilized thanks, above all, to vaccination and prevention.

## 6. Conclusions

Advances in proteomics have managed to relate several biomarkers to COVID-19, its severity, and the development of complications or prolonged COVID-19. This means better knowledge for the management of the disease, improving clinical care through risk stratification and treatments.

However, despite advances in research, challenges remain, especially with regard to the correlation between changes in protein profile and disease progression. Study results have sometimes been contradictory and inconclusive, as evidence, reproducibility, and clinical validation are often lacking, highlighting the need to continue investigating and refining these approaches to gain a more complete understanding of the pathophysiology of COVID-19 and improve therapeutic interventions.

## Figures and Tables

**Figure 1 ijms-26-00653-f001:**
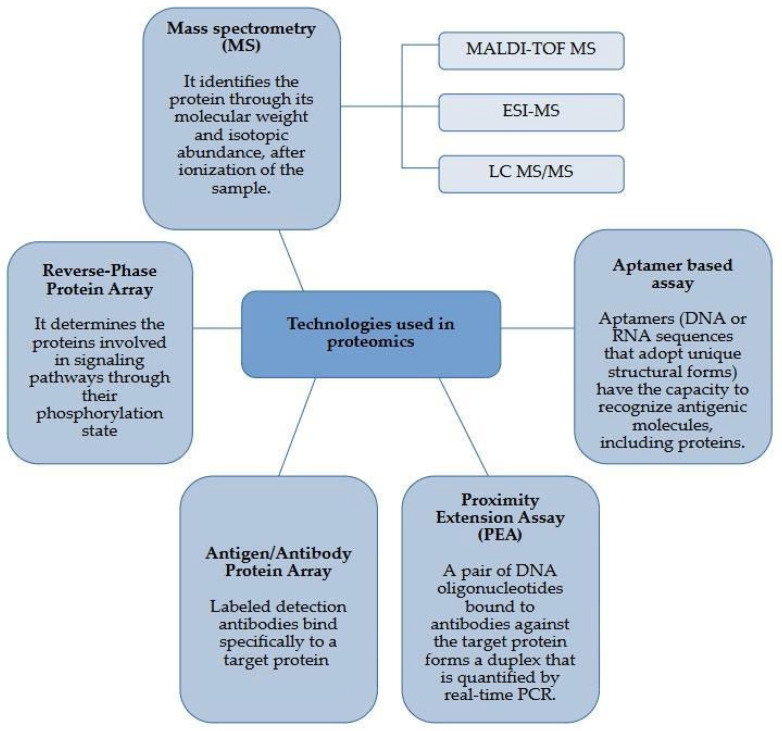
Technologies most used in proteomic analysis.

**Table 1 ijms-26-00653-t001:** Summary of key characteristics of invasive and non-invasive samples, as well as their clinical relevance in the context of COVID-19.

	Invasive Samples	Non-Invasive Samples
Types of samples	Blood and its components, cerebrospinal fluid, lung tissue, bronchoalveolar lavage andlavage and aqueous humor	Nasopharyngeal and oropharyngeal swabs, salive, urine and stools.
Advantages	Provide detailed information about the disease and its progression Allow analysis of specific proteins, viruses, and inflammatory markers	Lower risk of SARS-CoV-2 exposure. Easier to collect and less invasive for the patient. Useful in critical cases or in children.
Disadvantages	Requires invasive medical procedures Risk of infection or complications	Less detailed in some cases, may not reflect deep systemic changes.
Associated Markers	Blood: Inflammatory markers (IL-6, TNF-alfa), proteins related to thrombosis (D-dimer), endothelial damage markers (selectin E, von Villedbrand factor) Cerebrospinal fluid: Inflammatory markers in the CNS (IL-6. CXCL10), neurofilament, S100B, autoantibodies Lung Tissue: Alveolar damage markers (SP-A, SP-D), pulmonary fibrosis proteins (collagen types I and III, TFG-beta) Bronchoalveolar lavage: pulmonary inflammation (IL-6, TNF-alpha, CXCL10), proteins related to lung damage (CRP, surfactants) Aqueous Humor: Ocular damage markers (intraocular endothelial proteins, IL-6, IL-8)	Nasopharyngeal/Oropharyngeal Swabs: SARS-CoV-2 nucleocapsid and spike proteins, local inflammation markers (IL-6, TNF-α). Saliva: SARS-CoV-2-specific IgA, IL-1β, IL-8. Urine: Renal damage markers (NAG, NGAL), IL-6. Stool: Viral RNA, intestinal inflammation markers (IL-1β, TNF-α), changes in the gut microbiota.
Clinical Uses	Diagnosis of severe diseases or complications. Disease progression monitoring. Tissue damage and organ function assessment.	Initial diagnosis of COVID-19. Follow-up in asymptomatic or mild cases. Monitoring in high-demand scenarios without additional exposure risk.

**Table 2 ijms-26-00653-t002:** Summary of the biomarkers found in the literature during the year 2024 in relation to COVID-19.

Biomarkers	Predictive Value	Reference Values/Cut Points	Validation	Sample Type	References
Galectin 9 (LGALS9), Lysosomal Associated Membrane Protein 3 (LAMP3), Prostasin (PRSS8) and Agrin (AGRN)	Overexpression associated with severe disease	Continuous variable	No	Plasma	[82]
Threonine (Thr) and alanine (ALA)	Reduction in Thr in patients vs. controls. Reduction in ALA associated with greater disease severity	Continuous variable	No	Saliva	[67]
Serum amyloid A protein, tetranectin and neutrophil defensin 3	Indicators of atrioventricular block	Continuous variable	No	Plasma	[69]
PD-L1 soluble, TNF-R1 and IL-18BP	Overexpression indicative of secondary hemogaphocytic lymphohistiocytosis (sHLH)	sPD-L1 (8.44 NPX for sHLH vs. 7.8 for non-sHLH) TNF-R1 (9.46 NPX for sHLH vs. 8.53 for non-sHLH) IL-18P (8.07 NPX for sHLH vs. 7.35 for sHLH)	No	Plasma	[70]
Taurine	High levels are associated with a reduction in post-COVID syndrome	Continuous variable	No	Plasma	[83]
IL1RL1, elastin, Platelet Factor 4 (PF4) and SERPIN3	Association with fatal outcome after admission	Continuous variable	No	Plasma	[84]
GDF-15	Disease severity categorized into different levels	Mild: <0.0001 RFU/50 µL Moderate: <0.0001 RFU/50 µL Severe: 0.0049 RFU/50 µL Fatal: 0.0388 RFU/50 µL	Yes	Plasma	[74]
CCL7 and Carbonic Anhydrase (CA14)	Increased CCL7 and decreased CA14 as short-term predictors of symptom worsening	Continuous variable	Yes	Plasma	[85]
Cytokeratin-2e (K22E), Extracellular Matrix Protein (ECM1) and Alpha 2-antiplasmin (A2AP)	Diagnosis in early infection	Continuous variable	No	Plasma	[86]
PCR, ECM1, HECW1, ALS and LCAT	Increased CRP and decreased ECM1, HECW1, ALS and LCAT as early predictors of mortality	Continuous variable	No	Plasma	[23]
Cutaneous T cell-attracting chemokine (CTACK), M-CSF and IL-18	High levels associated with mortality	Continuous variable	No	Serum/bronchoalveolar lavage	[80]
IL-8, MCP-1, TNFRSF9, Delta/Notch-like EGF-related receptor (DNER), CCL4, Receptor-type tyrosine-protein kinase (Flt3L), CXCL10 and CD40	Overexpression associated with acute primary angle-closure glaucoma caused by COVID-19	Continuous variable	No	Aqueous humor	[22]
Dynein Cytoplasmic 1 (DYNC1) and Microtubule Associated Protein RP/EB Family Member 1 (MAPRE1)	Susceptibility to COVID-19 infection	Continuous variable	No	Saliva/plasma	[87]
Thrombospondin 1 (THBS1), Actinin Alpha 1 (ACTN1), Actin Alpha Cardiac Muscle 1 (ACTC1), POTE Ankyrin Domain Family Member F (POTEF), Actin Beta (ACTB), Tropomyosin 4 (TPM4), Vinculin (VCL), ICAM1	Dysregulated in Omicron infection	Continuous variable	No	Plasma	[88]
MSLN, FGF-5, CSF, EPHA2, TGF-α and β-NGF	High levels associated with serious illness	Continuous variable	Yes	Plasma	[89]
TRANC, FASLG, XPNPEP2 and CD207	Elevated in milder vs. severe COVID-19 disease. Association with better prognosis	Continuous variable	Yes	Plasma	[89]
PF4V1, NUCB1, CrkL, SerpinD1, Fen1, GATA-4, ProSAAS, PARK7 y NET1	Differentiation between COVID-19 and non-COVID-19 patients in ICU	Continuous variable	No	Plasma	[90]

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
