# Peer review of "Utility of Protein Markers in COVID-19 Patients"

_ijms, 2025, doi:10.3390/ijms26020653_

Round 1

Reviewer 1 Report

Comments and Suggestions for Authors

The manuscript is very general, not novel, and not a very well-rationalized proposal of biomarkers for COVID risk assessment and treatment. The authors tried to propose some general biomarkers based on proteomics. While the article's title states risk assessment, most results are post-mortem after severe disease and without stratifying previous medical conditions that will mislead the analysis of biomarkers.  There is a lot of text for a few defined Tables, which is very poor. I suggest that the authors rewrite the manuscript and do a real screening of the article since as is the paper is not suitable for publication.

Reviewer 2 Report

Comments and Suggestions for Authors

The manuscript "Use of Biomarkers in COVID-19 Risk Assessment and Treatment" by Lopez and colleagues describes a literature review on the topics of "SARS-CoV-2" and "COVID-19". The review is well written and collects some of the existing literature on COVID-19 biomarkers, with a preponderant focus on proteomic ones. While an interesting read, I feel the manuscript lacked in direction and overall supervision. Below, my points.

1) The title seems to aim at the discussion of biomarkers related to COVID-19 research. Yet, the search terms used by the authors, briefly presented in paragraph 2, are specifically aiming at "proteomics" and mass spectrometry. Then, the authors unsurprisingly find that protein markers are highly related to SARS-CoV-2 detection. I would suggest the authors to clearly present their review as a mass-spectrometry-focused biomarker review, and not with the terribly misleading "biomarkers" title (which could attract readers looking for clinical, nucleotididic, or metabolic biomarkers).

2) The search terms used in the literature search, as well as the database used, should be clearly stated. Sentences like "we used databases such as PubMed" (line 110) are too vague. The authors should list the entire list of peer-reviewed databases used, no "such as" examples. The same is true for the search terms. Which terms did the authors use? Again, stating that terms "such as 'proteomics'" were used (line 112) is overly vague and shows a too superficial and irreproducible approach for a scientific review.

3) Table 1: which of the presented markers are surprising, since they are not diretly related to SARS-CoV-2-triggered molecular pathways?

4) Also on Table 1: the authors should make an effort in adding evidence to each individual biomarker. The validity of any scientific finding is increased by independent evidence and reproducibility. None of the molecular biomarkers presented in Table 1 has more than one evidence in literature, which seems a bit odd and disconcerting. The authors should make an effort of finding multiple independent evidences for at least some of these biomarkers, otherwise it would resemble a glorified tabularization of a Google Scholar "best hit" research.

Round 2

Reviewer 1 Report

Comments and Suggestions for Authors

The authors have improved the manuscript. However, some points should be considered in this revised version. The first issue is the specificity of the protein biomarkers of the infection, which should be specific, and the text does not provide this element.  For example, the stratification performed in this article is simple, but useful  doi 10.1097/CCM.0000000000005983 

The second point is the relationship of specific markers; this article mentions important proteins like mesothelin https://doi.org/10.1016/j.heliyon.2023.e23320

while this article mentions the proteomic differences between COVID and long COVID https://doi.org/10.1186/s10020-023-00610-z and this one which refers to organ-specific markers

https://doi.org/10.1016/j.bbrep.2023.101493

Thus, the manuscript still needs improvement from that point of view. I hope the authors take into account the suggestions to improve the manuscript 

Round 3

Reviewer 1 Report

Comments and Suggestions for Authors

The authors have modified the manuscript according to the suggestions. The manuscript was improved. I have no further comments.